# An Algorithmic Approach Is Superior to the 99th Percentile Upper Reference Limits of High Sensitivity Troponin as a Threshold for Safe Discharge from the Emergency Department

**DOI:** 10.3390/medicina57101083

**Published:** 2021-10-12

**Authors:** Taekyung Kang, Gwang Sil Kim, Young Sup Byun, Jongwoo Kim, Sollip Kim, Jeonghyun Chang, Soo Jin Yoo

**Affiliations:** 1Emergency Medicine, Sanggye Paik Hospital, Inje University College of Medicine, Seoul 01757, Korea; emmania2011@paik.ac.kr; 2Internal Medicine, Sanggye Paik Hospital, Inje University College of Medicine, Seoul 01757, Korea; zidan007@paik.ac.kr (G.S.K.); ysbyun@paik.ac.kr (Y.S.B.); 3Health Promotion Center and Family Medicine, Sanggye Paik Hospital, Inje University College of Medicine, Seoul 01757, Korea; s2700@paik.ac.kr; 4Laboratory Medicine, Ilsan Paik Hospital, Inje University College of Medicine, Goyang 10380, Korea; lalacopine@gmail.com (S.K.); azacsss@naver.com (J.C.); 5Laboratory Medicine, Sanggye Paik Hospital, Inje University College of Medicine, Seoul 01757, Korea

**Keywords:** high-sensitivity cardiac troponin I, acute myocardial infarction, emergency department, chest pain

## Abstract

*Background and Objectives*: High-sensitivity cardiac troponin I (hs-TnI) is an important indicator of acute myocardial infarction (AMI) among patients presenting with chest discomfort at the emergency department (ED). We aimed to determine a reliable hs-TnI cut-off by comparing various values for a baseline single measurement and an algorithmic approach. *Materials and Methods*: We retrospectively reviewed the hs-TnI values of patients who presented to our ED with chest discomfort between June 2019 and June 2020. We evaluated the diagnostic accuracy of AMI with the Beckman Coulter Access hs-TnI assay by comparing the 99th percentile upper reference limits (URLs) based on the manufacturer’s claims, the newly designated URLs in the Korean population, and an algorithmic approach. *Results*: A total of 1296 patients who underwent hs-TnI testing in the ED were reviewed and 155 (12.0%) were diagnosed with AMI. With a single measurement, a baseline hs-TnI cut-off of 18.4 ng/L showed the best performance for the whole population with a sensitivity of 78.7%, specificity of 95.7%, negative predictive value (NPV) of 97.1%, and positive predictive value (PPV) of 71.3%. An algorithm using baseline and 2–3 h hs-TnI values showed an 100% sensitivity, 97.7% specificity, an NPV of 100%, and a PPV of 90.1%. This algorithm used a cut-off of <4 ng/L for a single measurement 3 h after symptom onset or an initial level of <5 ng/L and a change of <5 ng/L to rule a patient out, and a cut-off of ≥50 ng/L for a single measurement or a change of ≥20 ng/L to rule a patient in. *Conclusions*: The algorithmic approach using serial measurements could help differentiate AMI patients from patients who could be safely discharged from the ED, ensuring that patients were triaged accurately and did not undergo unnecessary testing. The cut-off values from previous studies in different countries were effective in the Korean population.

## 1. Introduction

Chest pain is a common symptom observed in clinical practice and a symptom that suggests acute myocardial infarction (AMI). Approximately 10% of all patients who visit the emergency department (ED) present with symptoms suggestive of AMI; however, 75–85% of them are found to have non-coronary diseases [1,2,3]. Therefore, quickly distinguishing AMI patients from other patients is a crucial step to reduce unnecessary examinations and hospitalizations, as well as to perform timely revascularization. 

Cardiac troponin (cTn) is a key indicator for early diagnosis, especially in non-ST segment elevation myocardial infarction (NSTEMI) without remarkable changes in the electrocardiogram (ECG) [4]. The improvements in the analytical performance of high-sensitivity cardiac troponin (hs-cTn) methods resulted in a paradigm shift for the diagnosis of a vast array of myocardial injuries, especially AMI [2]. Hs-cTn assays could detect very low levels of cTn and subtle changes with a reliable precision, enabling the rapid identification or ruling out of AMI in the ED [5,6]. 

Beckman Coulter (Beckman Coulter Inc., Brea, CA, USA) recently released a new version of the high-sensitivity troponin I (hs-TnI) assay called the Access hs-TnI assay, which was approved in 2018. In a previous study, we found that the Access hs-TnI assay met the performance criteria of hs-cTn assays, and we also reported that the 99th percentile upper reference limits (URLs) of the Korean population were lower than the manufacturer’s claims and differed according to sex and age [6,7].

The international guidelines recommend using the 99th percentile URL calculated using healthy subjects as a cut-off to diagnose myocardial injury [8]. However, some patients with a low level of cTn between the level of detection and the URL have a higher risk of AMI, so the URL may not be an appropriate threshold measurement to guide the safe discharge from the ED [9,10]. On the other hand, detecting low concentrations of cTn may cause confusion and unnecessary work-ups with false positive results. Therefore, some groups proposed various algorithms using the delta-cTn values as a means to distinguish patients who need further examination from those who can be safely discharged [11,12,13,14,15,16]. However, the 99th percentile URLs, the best hs-TnI cut-off value at presentation, and the absolute delta change values differ between reagents and manufacturers [13,17]. Moreover, there are differences in hs-TnI distributions in reference populations and the cut-offs in Asian populations differ from those in European populations [7,17]. The data on the clinical performance of the recently released Access hs-TnI assay are still limited, and it is not yet known whether the results reported in Europe or the United States will be applicable to Asian populations [12,15].

The purpose of this study was to evaluate the possibility of a rapid AMI diagnosis and safe discharge using the Beckman Coulter Access hs-TnI assay in Korean patients who visited the ED. We aimed to determine a safe and reliable diagnostic criteria by comparing the diagnostic accuracy using the 99th URLs based on the manufacturer’s claims, the newly designated URLs in Korean populations, and the algorithms provided by other researchers [12,15].

## 2. Materials and Methods

### 2.1. Study Design and Population

This study was a single-center retrospective observational study of patients who visited the ED of the Sanggye Paik Hospital between June 2019 and June 2020. The inclusion criteria were: (1) aged > 20 years; (2) presence of chest pain or equivalent ischemic symptoms (including epigastric or left upper quadrant pain or discomfort or pressure); and (3) measurement of hs-TnI levels at ED presentation. The exclusion criteria (based on medical records) were: (1) clear alternative cause for the suspected symptoms other than acute coronary syndrome (including trauma); (2) inter-hospital transfer for revascularization after initial blood tests; (3) discharge against medical advice without adequate work-up; (4) cardiac arrest or in-hospital death at the ED before adequate work-up; (5) history of coronary revascularization within 3 weeks; and (6) diagnosis of ST segment elevation myocardial infarction (STEMI) on presentation. The protocol was approved by the Institutional Review Board of Sanggye Paik Hospital, Seoul, Korea (SPIRB-2019-05-014) and the requirement for informed consent was waived due to the retrospective study design.

### 2.2. Data Collection and Clinical Assessment

Patient demographic information, risk factors, previous history of coronary artery disease, presenting symptoms, and time between symptom onset and ED visit, were extracted electronically from medical records. In addition, examination findings such as vital signs, ECG, laboratory results, coronary angiography, echocardiography, and radiologic findings, such as enhanced chest computed tomography (CT), thoracic aorta CT angiography, and pulmonary CT angiography, were collected. Final diagnoses were adjudicated retrospectively by a cardiologist and a physician in the ED by reviewing all available medical records pertaining to the patient from the time of ED presentation to their most recent follow-up of at least 6 months. 

AMI, an outcome indicator, was defined as recommended by the 4th universal definition guidelines [8]. Myocardial infarction (MI) was diagnosed when there was evidence of myocardial necrosis in relation to a clinical setting consistent with myocardial ischemia. Myocardial necrosis was diagnosed when there was a significant increase and/or decrease in at least one value of hs-TnI. AMI was categorized as Type 1 MI (primary coronary events) or Type 2 MI (ischemia due to increased demand or decreased supply, for example coronary spasm, tachyarrhythmia, or hypertensive crisis) according to the 4th universal definition guidelines [8]. The final diagnoses were classified into six categories: Type 1 MI; Type 2 MI; unstable angina; chronic coronary syndrome; non-coronary cardiac diseases; and non-cardiac disease. 

### 2.3. Clinical Performance Evaluations of Access hs-TnI

The hs-TnI values were measured with the Access hs-TnI assay on the UniCel DxI800 analyzer platform (Beckman Coulter) in the central laboratory of our hospital within an hour after blood was drawn. The hs-TnI results analyzed in this study included the baseline hs-TnI at the ER visit (0 h) and a second hs-TnI measurement performed two to three hours after the initial blood test (2–3 h). 

On the package insert of the Access hs-TnI assay, the manufacturer described a limit of detection of 2.3 ng/L and limit of quantification at 10% coefficient of variation (CV) of 5.6 ng/L. The manufacturer suggested an overall 99th percentile URL of 17.5 ng/L (men, 19.8 ng/L; women, 11.6 ng/L) with a corresponding CV of <10%. In our previous study, we noticed that our population showed a lower 99th percentile URL compared to the insert (9.5 ng/L in 600 healthy subjects). We also noticed that the 99th percentile URLs of hs-TnI were higher in men (men, 11.3 ng/L; women, 7.8 ng/L) and in the age group ≥50 years (≥50 years, 12 ng/L; <50 years, 8.5 ng/L). In this study, we compared the clinical performance of the 99th percentile URLs suggested by the manufacturer and calculated in our previous study. 

We used the concept of the current hs-cTn 0/2–3 h algorithms suggested by groups in Europe, Australia, and United States [15,16]. This algorithm classified patients into three groups (Rule-out, Observe, Rule-in) according to the absolute value of the baseline hs-TnI (0 h) and delta hs-TnI, the amount of change over time between the second measurement (2–3 h after the first blood test) and the baseline hs-TnI. To establish the optimal decision cut-offs in the algorithm, we attempted to find the values which maximized the positive predictive value (PPV) for the Rule-in and the negative predictive value (NPV) for the Rule-out. Patients who did not meet the Rule-out or Rule-in criteria were assigned to the Observe group requiring further evaluation. In addition, the number of patients assigned to each category and the ratio of AMI determined in each group were checked. 

### 2.4. Statistical Analysis

For the continuous variables, the median and interquartile range (IQR) were calculated, and a Student’s t-test or Mann–Whitney test was performed. For categorical variables, the frequency and percentage were calculated and a Chi square test or Fisher’s exact test was performed.

The sensitivity, specificity, PPV, and NPV were evaluated for the baseline hs-TnI values using the various 99th percentile URLs. Area under the curve (AUC) from a receiver operating characteristic (ROC) curve was calculated to compare the clinical performances of the different baseline hs-TnI cut-off values. Youden’s indexes (sensitivity + specificity − 1) were calculated to select the optimal cut-offs for initial hs-TnI values. 

To evaluate the safety of the algorithmic approach, the sensitivity and NPV were calculated in the Rule-out group, and the specificity and PPV were calculated in the Rule-in group for accuracy evaluation. In addition, the percentage of patients classified in the Rule-out group or the Rule-in group was calculated to evaluate the efficacy. Statistical analyses were performed using Analyse-it for Microsoft Excel 5.40.2 (Analyse-it software Ltd., Leeds, UK) and SPSS 18.0 (SPSS Inc., Chicago, IL, USA), and the significance level was set to *p* < 0.05.

## 3. Results

### 3.1. Characteristics of Study Population

A total of 1410 patients visited the ED with chest pain or the equivalent symptoms and underwent an hs-TnI test at presentation. Forty patients were excluded due to inter-hospital transfer for revascularization (N, 5); recent coronary revascularization (N, 4); discharge against medical advice without adequate work-up (N, 12); cardiac arrest in ED (N, 4) or in-hospital death within 24 h due to medical illness including cardiogenic shock, malignancies, or acute cerebrovascular events (N, 6); and multiple trauma including traffic accidents, chest contusion, and fall-down accidents (N, 9). Among 1370 patients, 74 patients who were diagnosed with STEMI were excluded from further analysis (74/1370, 5.4%). A total of 1296 patients were analyzed for the diagnostic performance of hs-TnI at the ED (Figure 1). The baseline characteristics of the patients are shown in Table 1. The mean age of the patients included in this study was 57.8 ± 16.5 years, and women (60.7 ± 16.6 years) were significantly older than men (55.4 ± 16.0 years) (*p* = 0.004). The frequencies of the risk factors such as diabetes, hypertension, and heart failure were higher in the AMI group than in the non-AMI group, although they did not reach the statistical significance values. The frequencies of previous coronary artery diseases (37.4% vs. 28.7%, *p* = 0.0485) and lower estimated glomerular filtration rates (eGFR, <60 mL/min/m^2^, 18.1% vs. 10.1%, *p* = 0.0045) were significantly higher in the AMI patients than in non-AMI patients.

### 3.2. Final Adjudicated Diagnosis

A total of 155 (12.0%) patients were diagnosed with AMI, including 107 (8.3%) patients with Type 1 MI and 48 (3.7%) with Type 2 MI. Other patients were grouped into non-MI (N = 1141, 88.0%), including unstable angina (51, 3.9%), chronic coronary syndromes (143, 11.0%), non-coronary cardiac diseases (148, 11.4%), and non-cardiac diseases (799, 61.7%). 

### 3.3. Concentrations of hs-TnI According to Final Diagnosis

The baseline Access hs-TnI levels were calculated according to their final adjudicated diagnosis, and the medians and IQRs were as follows: 96.5 ng/L (22.8–747.2) in Type 1 MI; 38.4 ng/L (13.3–124.8) in Type 2 MI; 5.0 (3.1–8.0) in unstable angina; 3.0 (2.3–5.4) in chronic coronary syndromes; 5.1 (3.0–9.8) in non-coronary cardiac diseases; and 3.0 (2.3–5.0) in non-cardiac diseases (Figure 2A). The Access hs-TnI levels at ED presentation were significantly higher in the AMI group (median 50.8 ng/L; IQR 21.2–421.9 ng/L) than in the non-AMI group (median 3.0 ng/L; IQR 2.3–6.0 ng/L) (*p* < 0.0001). 

Second follow-up hs-TnI levels were measured in 589 patients (45.4%) between 2–3 h after the first blood test (median 160 min). Delta hs-TnI levels were significantly higher in the AMI group (median 102.7 ng/L; IQR 20.8–444.5 ng/L) than in the non-AMI group (median 0.3 ng/L; IQR 0.0–1.0 ng/L; *p* < 0.0001). The delta-TnI levels were calculated according to the final adjudicated diagnosis, and the medians and IQRs were as follows: 144.3 ng/L (36.2–648.5) in Type 1 MI; 25.4 ng/L (4.0–134.7) in Type 2 MI; 1.0 (0.2–1.7) in unstable angina; 0.3 (0.0–1.0) in chronic coronary syndrome; 0.7 (0.0–1.0) in non-coronary cardiac diseases; and 0.3 (0.0–1.0) in non-cardiac diseases (Figure 2B).

### 3.4. Clinical Performance of Baseline Access hs-TnI Using Different Cut-Offs

The diagnostic performance of baseline Access hs-TnI levels for the diagnosis of AMI was measured using the AUC, and the results were as follows: 0.934 (95% CI, 0.910–0.958) in total; 0.921 (95% CI, 0.888–0.953) in men; 0.974 (95% CI, 0.958–0.991) in women; 0.930 (0.904–0.955) in those aged ≥50 years; and 0.928 (0.860–0.997) in those aged <50 years (Figure 3). The prime cut-off values using the Youden index from the ROC curves were: 18.4 ng/L in total; 18.8 ng/L in men; 17.5 ng/L in women; 16.8 ng/L in those aged ≥50 years; and 6.9 ng/L in those aged <50 years. The sensitivities, specificities, PPVs, and NPVs were evaluated using various cut-offs (Table 2). The 99th percentile URLs calculated in our population (in our previous studies) were lower than the URL claimed by the manufacturer, showing better sensitivities (*p* < 0.05) and NPVs (*p* > 0.05) but consistently poor specificities and PPVs (*p* < 0.001), both in the total population and all subgroups (men, women, ≥50 years, and <50 years) [7]. 

### 3.5. Validation of Access hs-TnI Algorithmic Approach

In the hs-TnI Access 0/2–3 h algorithm, 912 out of 1296 patients (70.4%) could be analyzed. There were 384 patients excluded from algorithm analysis, who could not be ruled in or ruled out using the initial hs-TnI values and had no adequate second hs-TnI measurement 2–3 h after the first blood test. The selected algorithm used a cut-off of <4 ng/L for a single measurement >3 h after the onset of symptom or an initial level of <5 ng/L and a change of <5 ng/L to rule a patient out, and a cut-off of ≥50 ng/L for a single measurement or a change of ≥20 ng/L to rule a patient in [15]. Among the 912 participants, 595 (65.2%) patients were classified as Rule-out, 141 (15.5%) were classified as Rule-in, and 176 (19.3%) were classified as Observe (Figure 4). Out of the 595 patients classified as Rule-out, none were diagnosed with AMI, and the sensitivity and NPV were 100%. Of the 141 subjects classified as Rule-in, 127 were diagnosed with AMI, representing a specificity of 97.7% and a PPV of 90.1%. Among the Observe group, 18 patients (18/176, 10.2%) were AMI patients. 

Among the patients in the Rule-out group and Observe group, 55.1% (425/771) were followed up for longer than one month (Figure 4). The 30-day mortality rate was 0% and one patient in the Rule-out group was diagnosed with NSTEMI within a month. His first ER visit was 4 days after the symptom onset and he had an hs-TnI level of <2.3 ng/L and normal ECG findings, so he was discharged without a second hs-TnI measurement.

An internal validation was performed to confirm the reproducibility of the algorithmic approach. The total patients (N = 1296) were divided into derivation (60%) and validation (40%) cohorts according to the chronological order of ED visits. After excluding the patients who could not be classified into Rule-in or Rule-out with initial hs-TnI and who did not have an adequate second hs-TnI measurement, 518 and 394 patients, respectively, were included in the derivation and validation cohorts (Figure 5). The difference in the proportions of patients, PPV, and the specificity was not statistically significant. 

## 4. Discussion

The recent improvements in the analytical performance of hs-cTn assays enabled the detection of low levels of cTn and subtle changes in cTn and these assays played an important role in the clinical judgement for AMI in the ED. This study was performed to evaluate the diagnostic accuracy and clinical usefulness of the Access hs-TnI assay as an early screening tool for the detection of AMI. 

Heart failure and low eGFR are known to increase cTn values [2,5]. In our population, the AMI patients showed the higher frequencies of heart failure (5.8% vs. 3.0%, *p* = 0.1133), and the lower eGFR <60 mL/min/1.73 m^2^ (18.1% vs. 10.1%, *p* = 0.0045) (Table 1). The patients with low eGFR showed higher initial hs-TnI values in both non-AMI patients (median 10.5 ng/L; IQR 6.0–17.3 ng/L in low eGFR group vs. median 2.3 ng/L; IQR 2.3–3.0 ng/L in normal eGFR group) and in AMI patients (median 45.0 ng/L, IQR 35–1088 ng/L in low eGFR group vs. median 46.5 ng/L, IQR 8.8–229.3 ng/L in normal eGFR group). Physicians need to be cautious when making clinical decisions using hs-TnI values in patients with low eGFR, in which case the delta values of hs-TnI can be particularly helpful.

In the previous study, we set the 99th percentile URL in our population according to international guidelines [7,8]. In this study, we evaluated the clinical performance of these 99th percentile URLs. The lower URLs used in our populations did not provide definite benefits compared to the manufacturer’s URLs because improvements in the NPV did not reach a statistical significance, but the PPV was significantly poorer in the total population and all subgroups (Table 2). The predictive values depend on the prevalence, so the low prevalence of AMI in our population may result in poor predictive values.

The prime cut-off values calculated from the ROC curves were not significantly different in men (18.8 ng/L) and women (17.5 ng/L) with proximities to the manufacturer’s 99th percentile URL (17.5 ng/L in total population). However, the best cut-off values from the ROC curves were significantly lower in patients <50 years (6.9 ng/L) compared to patients ≥50 years (16.8 ng/L). This finding was the same when we divided the groups with a cut-off of 65 years (5.3 ng/L in patients <65 years and 21.1 ng/L in patients ≥65 years, data not shown). The previous guidelines emphasized the sex-specific 95th percentile URLs, but these data suggested that it may be necessary to consider URLs based on age groups [18,19]. This was consistent with the difference in 99th percentile URLs by age group, as found in our previous study [7,20].

To overcome the shortcomings of the diagnostic performance with a single cut-off, many algorithmic approaches were proposed. Table 3 summarizes the published data for the algorithmic approaches using hs-TnI assays in more than 500 participants. There were two published studies on algorithmic approaches using the Access hs-TnI assays. Boeddinghaus et al. evaluated the clinical performance of a 0/1 h algorithm in 1579 patients [12]. In the validation cohort of this study (N, 680), 60% of patients were ruled out (NPV, 99.8%), and 14% of patients were ruled in (PPV 73.9%). The 30-day survival rate in the Rule-out group was 100%. Nestelberger et al. verified the 0/2 h algorithm in 1280 subjects, reporting that 77.9% of the patients were in the Rule-out group (99.8% of NPV) and 5.8% were in the Rule-in group (77.0% of PPV) [15]. 

We were also able to confirm that the algorithmic approach resulted in a higher accuracy (100% sensitivity, 97.7% specificity) and more reliable predictabilities (100% NPV, 90.1% PPV) than the single measurements of any of the cut-offs (sensitivity 78.7–83.9%, specificity 86.8–95.7%, PPV 46.3–71.3%, NPV 97.1–97.5% in the overall population; Table 2). These results were superior to those of previous studies undertaken in Western countries [11,12,15,16]. The high PPV (90.1%) in this algorithm allowed for the selective provision of appropriate procedures in patients with ischemic symptoms. At the same time, the high NPV (100%) allowed for the appropriate safe discharge of selective patients following triage in the ED, avoiding unnecessary angiographies. This algorithm could reduce the length of time patients spent in the ED, and alleviate the unnecessary additional blood collection process, thereby allowing a quick discharge. It could also help to avoid overcrowding in the ED.

This algorithm showed a high efficiency considering that 80.7% of all the suspected patients were classified into a Rule-in group or a Rule-out group, and only 19.3% were classified into an Observe group. In addition, even with the first hs-TnI assay alone, 60.5% of all the patients were classified into Rule-in (9.8%) or Rule-out (50.7%) groups without the need for the second cTn test, so this algorithm was useful for establishing a rapid decision-making strategy.

Nevertheless, when applying the 0/2–3 h algorithm using hs-TnI levels in clinical practice, the following should be noted. In this study, 176 (19.3%) patients were in the Observe group, and 18 (10.2%) of these patients were diagnosed with AMI. Separating the Observe group enabled the selection of patients who needed further work-ups and avoided risky discharge, even if hospitalization and rapid revascularization were not required. Other studies reported that the Observe group patients had lower 30-day mortality rates, but similar 1-year mortality rates, compared to the Rule-in group patients [13]. Therefore, the symptomatic Observe group needed a careful clinical observation in the ED and the administration of appropriate treatment for each individual, considering their underlying diseases. In some cases, coronary angiography, cardiac CT, and additional ECG testing would be necessary, as well as treatment for underlying diseases such as arrhythmia, heart failure, and hypertension.

In particular, it is known that a single measurement requires the careful consideration of the time elapsed since symptom onset [1,21]. In the ED, a single low hs-TnI value is useful to rule out patients with a symptom duration longer than 3 h. However, if the the initial test value is very low in patients with a symptom onset within 3 h, the 0/1 h or 0/2–3 h algorithm would be safer despite the high NPV. In our study population, one patient who showed a low initial hs-TnI value <2.3 ng/L was discharged without a second measurement and actually revisited the ED and was diagnosed with NSTEMI.

Furthermore, it should be noted that 90.1% of the patients assigned to the Rule-in group were patients with AMI, and they required urgent percutaneous coronary intervention. In addition, even if type 1 MI was not diagnosed in the Rule-in group, those patients presented with unstable conditions requiring hospitalization, such as heart failure, myocarditis, unstable angina, acute cholecystitis, acute pyelonephritis, and pulmonary embolism. Therefore, in order to differentiate between these diseases, the coronary angiography for diagnosis and treatment could be considered for patients in the Rule-in group.

This study had several limitations. First, as a retrospective study, this study included patients who underwent hs-TnI on ED visits, and some patients who did not undergo a second hs-TnI test within 2–3 h were excluded. If the hs-TnI test was performed late or omitted based on the judgment of the medical staff, the patient was excluded from this algorithmic analysis. This could lead to selection bias and the omission of some cases of AMI. Second, it was difficult to guarantee the long-term safety of the Rule-out or Observe group in the 0/2–3 h algorithm because 44.9% (346/771) of patients did not visit our outpatient clinic after one month and a long-term follow-up was not performed in this study. Further prospective studies are required to compensate for these weaknesses and to verify the outcomes. Third, since the patients were recruited from a single regional ED, the local characteristics of the institute could impact the results. A multi-center prospective study is recommended to verify the accuracy and usefulness of the hs-TnI test and this algorithmic approach. Fourth, it is desirable to validate the analysis performance of the algorithm in each gender and in patients with renal impairments. However, in this study, subgroup analysis was not possible as the number of AMI was small in women (N, 39) and the low eGFR group (N, 29). In the future, we plan to conduct additional subgroup analyses through a prospective study with more patients. 

Nevertheless, this study was meaningful as it aimed to assess the clinical performances of Access hs-TnI in Koreans, whereas most of the published data were from one multinational study group or from Western countries. This study was the first to attempt an algorithmic approach for the diagnosis of AMI using the Access hs-TnI assay in an Asian population, involving the independent evaluation of the 99th URLs within the same institution.

## 5. Conclusions

In summary, the algorithmic approach, rather than the use of single hs-TnI values, was a more efficient measurement for the early diagnosis of AMI and safe discharge from the ED. We found that the cut-off values used in the algorithmic approaches from previous studies in different countries were effective in the Korean population.

## Figures and Tables

**Figure 1 medicina-57-01083-f001:**
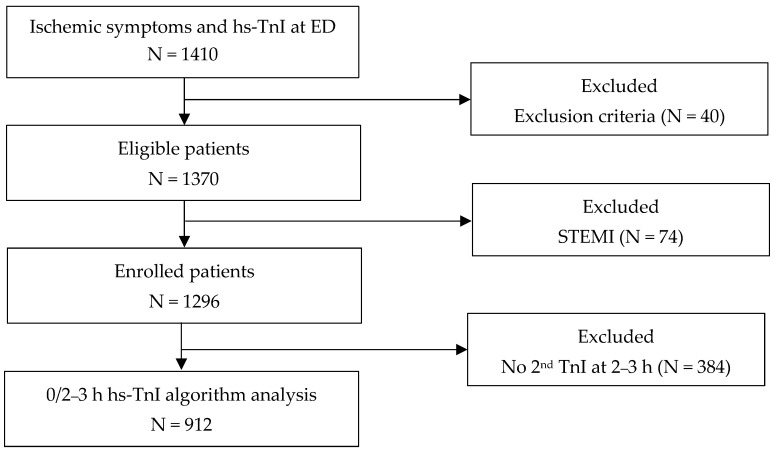
Patient enrollment process.

**Figure 2 medicina-57-01083-f002:**
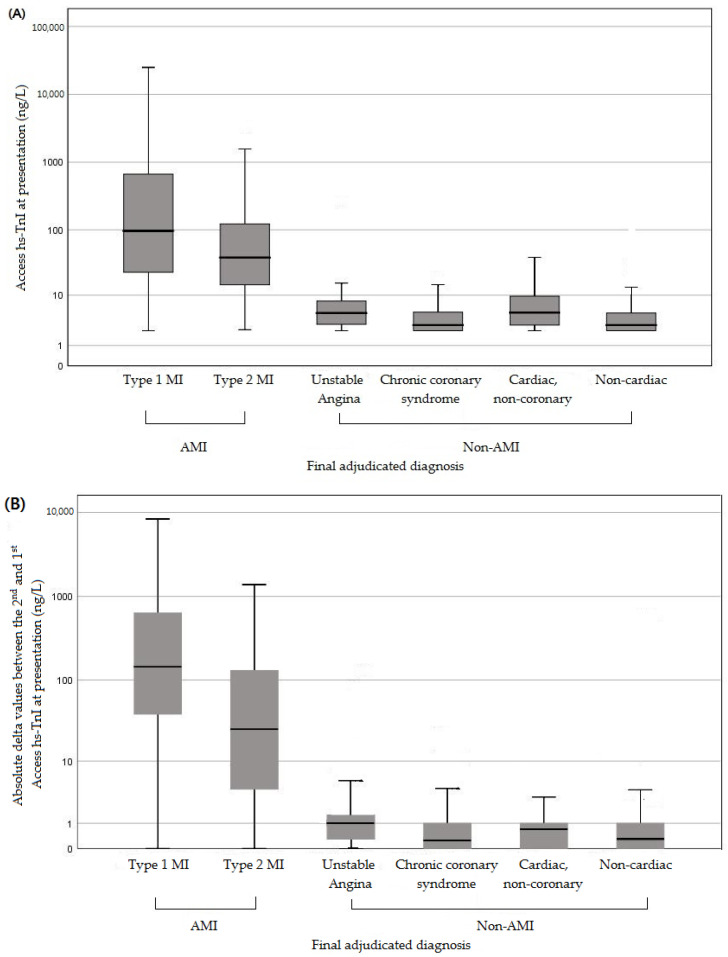
Boxplots of baseline at presentation (**A**) and absolute delta values (**B**) for the second and the first measurements with Access hs-TnI levels according to the final adjudicated diagnosis. The boxes represent medians and IQRs, and the whiskers represent the smallest and the largest non-outliers.

**Figure 3 medicina-57-01083-f003:**
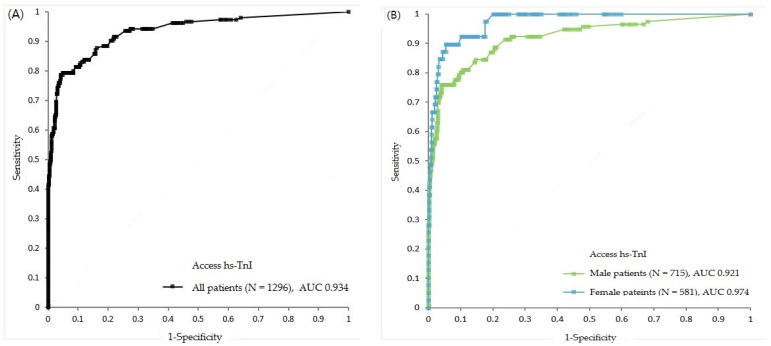
Receiver operating characteristic curves presenting diagnostic performance of Access hs-TnI assay at presentation for the diagnosis of AMI in all patients (**A**) and in each gender: men and women (**B**).

**Figure 4 medicina-57-01083-f004:**
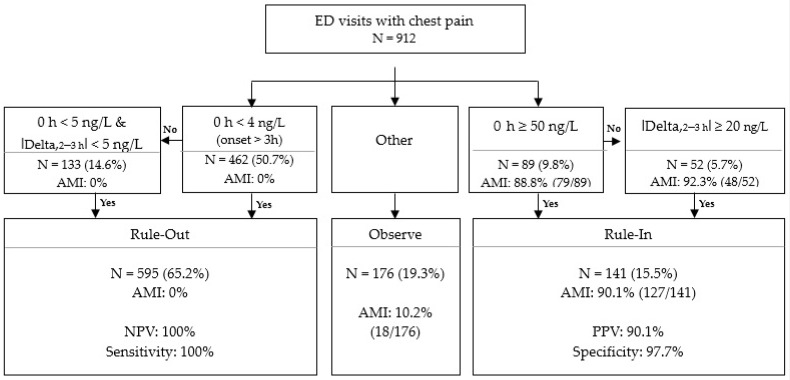
0/2–3 h algorithmic approach for AMI diagnosis using the baseline Access hs-TnI values at ED presentation and the absolute delta values between the second and baseline hs-TnI.

**Figure 5 medicina-57-01083-f005:**
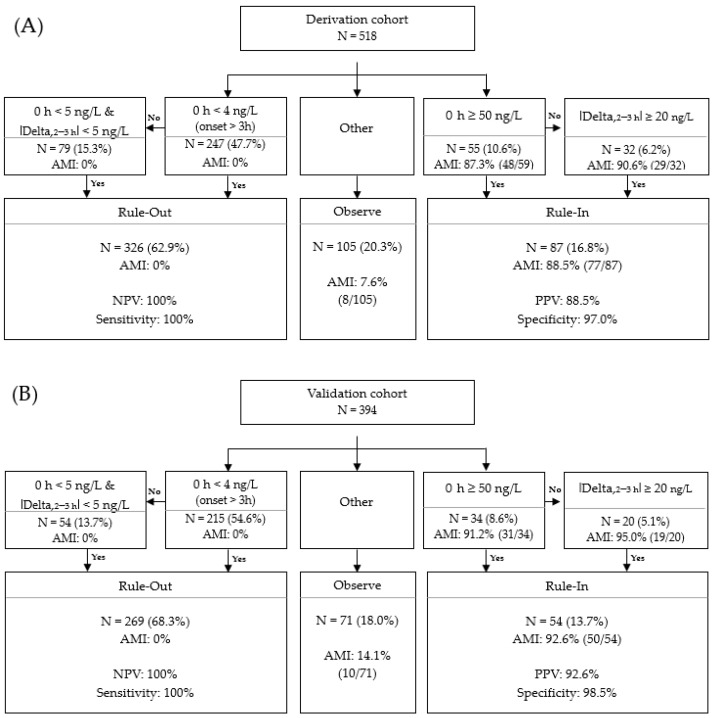
Internal validation with the derivation cohort (**A**) and validation cohort (**B**) regarding the 0/2–3 h algorithmic approach for AMI diagnosis using Access hs-TnI values at ED.

**Table 1 medicina-57-01083-t001:** Baseline characteristics and Access hs-TnI levels of 1296 patients with ischemic symptoms.

	All Patients	AMI	Non-AMI	*p*-Values
Number of patients–N (%)	1296	155 (12.0%)	1141 (88.0%)	
Age, years–median (IQR)	59.0 (25–69)	62.0 (52–70)	58.0 (46–69)	0.0008
Female gender–N (%)	581 (44.8%)	39 (25.2%)	542 (47.5%)	<0.0001
Early presenters (<3 h from onset)–N (%)	510 (39.4%)	82 (52.9%)	428 (37.5%)	0.0008
History and risk factors–N (%)				
Coronary artery disease	385 (29.7%)	58 (37.4%)	327 (28.7%)	0.0485
Diabetes	228 (17.6%)	37 (23.9%)	191 (16.7%)	0.0540
Hypertension	508 (39.2%)	71 (45.8%)	437 (38.3%)	0.1123
Heart failure	43 (3.3%)	9 (5.8%)	34 (3.0%)	0.1133
Chronic kidney disease	36 (2.8%)	5 (3.2%)	31 (2.7%)	0.9549
eGFR < 60 mL/min/1.73 m^2^–N (%)	143 (11.0%)	28 (18.1%)	115 (10.1%)	0.0045
Baseline hs-TnI, ng/L–median (IQR)	3.5 (2.3–8.0)	50.8 (21.2–421.9)	3.0 (2.3–6.0)	<0.0001

*p*-values for the difference between the AMI and non-AMI groups. Abbreviations: AMI, acute myocardial infarction including type 1 and 2; IQR, interquartile range; eGFR, estimated glomerular filtration rate measured using CKD-EPI equations.

**Table 2 medicina-57-01083-t002:** Diagnostic performance of baseline Access hs-TnI using various cut-offs for AMI diagnosis.

Group	Cut–Offs	Sensitivity	NPV	Specificity	PPV
Overall	99th URL by manufacturer, 17.5 ng/L	79.4 (72.1–85.4)	97.1 (96.1–97.9)	95.0 (93.6–96.2)	68.3 (62.4–73.9)
	99th URL in our population, 9.5 ng/L	83.9 (77.1–89.3)	97.5 (96.5–98.3)	86.8 (84.7–88.7)	46.3 (42.3–50.5)
	Cut-off from ROC in this study *, 18.4 ng/L	78.7 (71.4–84.9)	97.1 (96.1–97.8)	95.7 (94.4–96.8)	71.3 (95.3–97.9)
Men	99th URL by manufacturer, 19.8 ng/L	73.3 (64.3–81.1)	94.9 (93.2–96.2)	96.0 (94.1–97.4)	78.0 (70.2–84.2)
	99th URL in our population, 11.3 ng/L	78.5 (69.8–85.5)	95.6 (93.9–96.9)	90.5 (87.9–92.7)	61.5 (55.0–67.5)
	Cut-off from ROC in this study, 18.8 ng/L	75.9 (67.0–83.8)	95.4 (93.7–96.6)	96.0 (94.1–97.4)	78.7 (70.9–84.6)
Women	99th URL by manufacturer, 11.6 ng/L	92.3 (79.1–89.4)	99.4 (98.2–99.8)	88.6 (85.6–91.1)	36.7 (31.1–42.7)
	99th URL in our population, 7.8 ng/L	100 (91.0–100)	100	79.5 (75.9–82.8)	26.0 (22.9–29.3)
	Cut-off from ROC in this study, 17.5 ng/L	89.7 (75.8–97.1)	99.2 (98.1–99.7)	94.6 (92.4–96.4)	54.7 (45.4–65.6)
≥50 years	99th URL, overall, by manufacturer, 17.5 ng/L	80.0 (71.9–86.6)	96.7 (95.4–97.7)	93.3 (91.4–95.0)	65.4 (58.9–71.3)
	99th URL in our population, 12.0 ng/L	81.6 (73.7–88.0)	96.7 (95.4–97.7)	86.3 (83.7–88.6)	48.3 (43.5–53.1)
	Cut-off from ROC in this study, 16.8 ng/L	80.0 (71.9–86.6)	96.7 (95.4–97.7)	92.6 (90.5–94.3)	62.9 (56.6–68.7)
<50 years	99th URL, overall, by manufacturer, 17.5 ng/L	76.7 (57.7–90.0)	98.0 (94.3–99.0)	98.9 (97.1–99.7)	85.2 (67.9–93.9)
	99th URL in our population, 8.5 ng/L	83.3 (65.3–94.4)	98.5 (96.8–99.3)	95.4 (92.6–97.4)	61.0 (48.4–72.0)
	Cut-off from ROC in this study, 6.9 ng/L	86.7 (69.3–96.2)	98.8 (97.0–99.5)	82.8 (89.9–95.4)	51.0 (41.6–61.4)

Numbers in parenthesis mean 95% confidence intervals. * Youden index was calculated as follow; sensitivity + specificity − 1.

**Table 3 medicina-57-01083-t003:** Published data for algorithmic approaches using high-sensitivity troponin I assays.

Study	Boeddinghaus et al. [12]	Nestelberger et al. [15]	Boeddinghous et al. [11]	Nowak et al. [16]		This Study
High-sensitivity troponin I assays	Beckman, Access hs-TnI	Beckman, Access hs-TnI	Quidel, TrageTrue hs-TnI	Siemens, Atellica hs-cTn		Beckman, Access hs-TnI
Study population	APACE study *	APACE study *	APACE study *	29 US medical centers		One Korean center
No. of participants	680	1280	545	2113	1916	912
AMI cases, %	15.4	6.9	14.0	11.8	11.8	12.0
Algorithm	0/1 h	0/2 h	0/1 h	0/1 h	0/2–3 h	0/2-3 h
Rule-out cut-off						
0 h hs-TnI for direct rule out, ng/L	4	4	4	3	3	4
0 h hs-TnI for algorithmic rule out, ng/L	5	5	5	6	7	5
Delta hs-TnI	4	5	3	3	7	5
Rule-in cut-off						
Baseline hs-TnI, ng/L	50	50	60	120	120	50
Delta hs-TnI, ng/L	15	20	8	12	20	20
Rule-out, %	60.0	77.9	55.0	50.4	55.6	66.1
Rule-in, %	14.0	5.8	18.0	12.6	13.3	15.5
Sensitivity, %	98.9	97.7	100	98.7	99.1	100
NPV, %	99.8	99.8	100	99.7	99.9	100
Specificity, %	95.9	98.6	95	95.7	95.5	97.7
PPV, %	73.9	77.0	76.8	69.4	69.7	90.1

* APACE (Advantageous predictors of acute coronary syndrome evaluation) is an international multicenter study with 12 centers in 5 countries (Australia, New Zealand, Spain, Switzerland, and United States).

## Data Availability

Not applicable.

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
