# Peer review of "An Algorithmic Approach Is Superior to the 99th Percentile Upper Reference Limits of High Sensitivity Troponin as a Threshold for Safe Discharge from the Emergency Department"

_medicina, 2021, doi:10.3390/medicina57101083_

Round 1

Reviewer 1 Report

The authors describe the 99th cut-offs of the Beckman Coulter Access hs-cTnI

and derive a rapid triage algorithm for acute myocardial infarction in a Korean chest pain cohort of nearly 1300 patients included in 1 year in a single center, retrospective study. The manuscript is well written and shows clearly the recommended clinical decision values (CDV`s or URL`s) by the manufacturer are not perfectly applicable to all.

Major comments:

At what time point was the follow-up performed for the adjudication of the final diagnosis? The timing and available information at this time point might result different adjudicated diagnosis.

Is there any MACE criteria available at this adjudication time point? Incorporating this information could enhance the safety of the derived algorithm.

Please explain in the methods section how and according to which statistical criteria the algorithm was developed (to maximise NPV and PPV for example). Additionally, to increase the stability and reproducibility of your results, I would recommend to divide the population into derivation and validation cohort for the algorithm development (60:40 or 70:30 for example).

It is recommended to perform a subgroup analysis to check how the algorithm performed among men and women and in those with impaired kidney function (eGFR <60)

Interestingly, the derived algorithm is almost the same like the ESC 0/2h algorithm except the direct rule-out criteria in late presenter criteria (>3h CPO) with the very low hs-cTnI concentrations. In your algorithm it is 4ng/L as well, but apparently not in late presenters (even though mentioned in line 321?). Does the direct rule-out criteria change if only applied in late presenters?

Since the algorithms are almost the same for this assay in different derived populations, maybe these patient cohorts are not so different after all. This should be discussed.

Minor comments:

Introduction

Line 48: the word “perform” should be inserted before timely revascularization, otherwise it reads as revascularization should be reduced.

The references 1 and 4 (Roffi 2016 and Hamm 2012) are old; the new ESC guidelines as published in 2020 should be used instead.

Methods

Line 91: 40 patients were retrospectively excluded because of a clear alternative cause. It might be useful to indicate what was the diagnosis in these patients as this may introduce a bias in the assessed population because it could generate a different population with different derived CDV`s. It needs to be clear that even prospectively included, many of those patients would not have been included and therefore the study population would be the same.

Line 151: Was the Youden index used for derivation of the best cut-off from the AUC?

Results/discussion:

Line 206: 589 patients were available for a 2nd hs-cTnI measured between 2-3h but in the figure it is 912 patients available for the algorithm development? What was the median time to the 2nd measurement?

In table 4, the delta for rule-in is 15ng/L but in the figure it is 20ng/L.

The single cutoff approach with the previously derived (lower) URL`s showed a higher sensitivity and NPV than in this study in all, among gender etc except in those <50 years of age. How do the authors explain these findings: did the populations differ substantially, was a different methodology used?

In terms of rule-in with the single cut-off approach there was a comparable specificity and PPV for the URL`s derived in this study as compared to the manufacturers URL`s and a worse performance of the ones previously derived (except in <50y), this should be discussed.

Author Response

Responses to the comments from the reviewers:

To reviewer 1
Thank you very much for your thoughtful suggestions and insights. The manuscript has benefited from these insightful suggestions. The manuscript has been rechecked and the necessary changes have been made in accordance with the reviewers’ suggestions. Changes in the main text have been highlighted in yellow.

Major comments:

1. At what time point was the follow-up performed for the adjudication of the final diagnosis? The timing and available information at this time point might result different adjudicated diagnosis.

⇒ Thank you for your insightful comment. In the case of the ‘Rule in’ group, most patients were hospitalized via the emergency room and coronary angiography or cardiac CT was performed within 3 days. Thus, the final diagnosis was made within 3 days in most cases. In the ‘Observe group’, additional examinations such as cardiac CT or coronary angiography was determined based on the patient’s symptom or the interval change of ECG. Most ‘Rule out’ patients were followed up in an outpatient center. Similarly, additional tests were performed at the discretion of the specialist or just follow-up or ended the treatment. For coronary angiography, the longest interval was 55 days from the emergency room visit. We determined the final diagnosis by analyzing the electronic medical record. This study was a retrospective study based on the chart review, and the final diagnosis for the ‘Observe group’ and ‘Rule out group’ were analyzed using all available medical records pertaining to the patient from the time of ED presentation to their most recent follow-up at least 6 months. In lines 108-109, we added the words ‘retrospectively’ and ‘at least 6 months'.

2. Is there any MACE criteria available at this adjudication time point? Incorporating this information could enhance the safety of the derived algorithm.

 Your comment is a very important point. People hospitalized for AMI or heart failure have a MACE score on EMR, and the MACE criteria were used to classify these patients, but this was not a requirement for this study. Since this study was aimed to evaluate the diagnostic accuracy of hs-TnI for safe discharge from the emergency department, we did not include the MACE score in this analysis. We will make sure to include the MACE criteria in future further study.

3. Please explain in the methods section how and according to which statistical criteria the algorithm was developed (to maximize NPV and PPV for example). Additionally, to increase the stability and reproducibility of your results, I would recommend to divide the population into derivation and validation cohorts for the algorithm development (60:40 or 70:30 for example).

 In the ‘2.3. Clinical performance evaluations of Access hs-TnI’, we added a sentence as below (line 141).

“To establish the optimal decision cut-offs in the algorithm, we tried to find the values that maximize the PPV for rule-in and the NPV for rule-out.”

As you recommend, verification with a separate derivation cohort and a validation cohort is ideal. Due to the limitation of time and available patients groups, we could not validate with new independent patients groups. So we used the internal validation method as you recommend. Total patients (N = 1,296) were divided 60:40 ratio along the sequence of ED visits and finally, the derivation and the validation cohort included 518 and 394 patients, respectively. Their results were added in the final parts of the ‘Results” section with figure 5-(A) and –(B). Results of each algorithm (total population, derivation cohort and validation cohort) were not significantly different. It was described in lines 266-272.

4. It is recommended to perform a subgroup analysis to check how the algorithm performed among men and women and in those with impaired kidney function (eGFR <60)

⇒ There was a total of 581 women (44.8%) in our group, but 39 of them had AMI, so subgroup analysis was impossible. In addition, in our group, eGFR < 60 was a total of 143 patients, among which 29 patients had AMI, so subgroup analysis was also impossible. However, for the low eGFR, the frequencies and hs-TnI values in each AMI and non-AMI group were described in table 1 and in the second paragraph of ‘Discussion’. 

5. Interestingly, the derived algorithm is almost the same as the ESC 0/2h algorithm except the direct rule-out criteria in late presenter criteria (>3h CPO) with the very low hs-cTnI concentrations. In your algorithm it is 4ng/L as well, but apparently not in late presenters (even though mentioned in line 321?). Does the direct rule-out criteria change if only applied in late presenters? Since the algorithms are almost the same for this assay in different derived populations, maybe these patient cohorts are not so different after all. This should be discussed.

We have revised our algorithm and data because we thought your comments were critical. Those who performed initial TnI tests < 3 h of onset could not be ruled out only with initial TnI values in the revised data. In the previous data, of the patients who were excluded due to initial TnI <4, a hundred patients had initial TnI values within 3h after the symptom onset. Among them, 92 patients were allocated to ‘Rule-out’ group because they had initial TnI <5 and delta TnI <5. Nine patients among them were reassigned to the ‘Observe group’. Accordingly, all of the N (numbers of patients) and % values were corrected in figure 4 and the manuscript, and they were written in yellow highlights. Since there were no MI patients among them, NPV and sensitivity did not change. In the abstract, '3 hours after symptom onset' was added to line 31.

 Minor comments:

 Introduction

1. Line 48: the word “perform” should be inserted before timely revascularization, otherwise it reads as revascularization should be reduced.

 ⇒ We added ‘to perform’ in that sentence (line 49). 

 2. The references 1 and 4 (Roffi 2016 and Hamm 2012) are old; the new ESC guidelines as published in 2020 should be used instead.

 ⇒ We deleted reference 4 and replaced with the new ESC guideline 2020 as a new reference 4. 

 Methods

3. Line 91: 40 patients were retrospectively excluded because of a clear alternative cause. It might be useful to indicate what was the diagnosis in these patients as this may introduce a bias in the assessed population because it could generate a different population with different derived CDV`s. It needs to be clear that even prospectively included, many of those patients would not have been included and therefore the study population would be the same.

 ⇒ We added numbers of patients corresponding to each, and some details of patients’ states as below (in line 167-174). 

“Forty patients were excluded due to inter-hospital transfer for revascularization (N, 5), recent coronary revascularization (N, 4), discharge against medical device without adequate work-up (N, 12), cardiac arrest in ED (N, 4) or in-hospital death within 24 hours due to medical illness including cardiogenic shock, malignancies, or acute cerebrovascular events (N, 6), and multiple trauma including traffic accidents, chest contusion, and fall-down accidents (N, 9).”

 4. Line 151: Was the Youden index used for derivation of the best cut-off from the AUC?

 ⇒ We used the Youden index as you mentioned. 

We added a sentence in the ‘2.4. Statistical analysis’section, as below (line 155)

Youden’s indexes (sensitivity + specificity - 1) were calculated to select the optimal cut-offs for initial hs-TnI values.”

And the following sentence was added at the bottom of Table 3. 

*Youden index was calculated as follows; sensitivity + specificity -1.”

Results/discussion:

5. Line 206: 589 patients were available for a 2nd hs-cTnI measured between 2-3h but in the figure, it is 912 patients available for the algorithm development? What was the median time to the 2nd measurement?

 ⇒ 589 patients who had second hs-TnI values were included in the algorithm analysis. In addition, 323 patients were also included in the algorithm analysis, because they could be allocated into rule-out or rule-in group due to high (>50) or low (<4) initial hs-TnI values. And, the median time of second hsTnI measurements was 160 minutes (2h40min). The median time was inserted in the sentence of “3.3. Concentrations of hs-TnI according to final diagnosis” as below (line 215).

“Second follow-up hs-TnI levels were measured in 589 patients (45.4%) between 2-3 h after the first blood test (median 160 minutes).” 

In addition, we added some sentences about the numbers of patients included in the algorithm (in line 249-251).

 6. In table 4, the delta for rule-in is 15ng/L but in the figure it is 20ng/L. 

 ⇒ It was the clerical error. We have corrected “15 ng/L” to “20 ng/L” in table 4.

7. The single cutoff approach with the previously derived (lower) URL`s showed a higher sensitivity and NPV than in this study in all, among gender etc except in those <50 years of age. How do the authors explain these findings: did the populations differ substantially, was a different methodology used?

In terms of rule-in with the single cut-off approach there was a comparable specificity and PPV for the URL`s derived in this study as compared to the manufacturers URL`s and a worse performance of the ones previously derived (except in <50y), this should be discussed.

 ⇒ I agree with your point, but in the current state, it is not possible to accurately explain the causes of the low performance for a single cut-off in this study. We have reviewed several references, but it was difficult to make a consistent comparison due to the difference in the participants, the elapsed time from symptom onset to the hospital visit, and the hs-TnI reagents. Among them, the study by Christenson et al* was comparable since it used the same reagent and present the URLs of Access hs-TnI reagent inserts. 

(*Christenson R.H. et alAnalytical and clinical characterization of a novel high-sensitivity cardiac troponin assay in a United States populationClinical Biochemistry 2020, 83, 28-36)

 We found that the sensitivity and NPV of our patient group were lower, and the specificity and PPV were better than those in the study by Christenson et al. This means that the hs-TnI value in our group is lower because the distributions of patients' age, gender, and collection time after the symptom onset id not differ significantly between two studies. Among AMI patients in our group, 32 patients (20.6%) showed false negative because the initial hs-TnI was lower than the insert cut-off 17.5 ng/L and their median time from symptom onset to ED presentation was 1.0 h. On the other hand, in 133 patients who showed true positive for insert cut-off 17.5, their median time from symptom to ER visit was 3.8h. It suggests the time window from symptom onset to the ED visits and to the blood sampling plays an important role, but it was difficult to show adequate evidence in this study, and further prospective research is likely to be needed.

 The cut-offs derived from this study are not 99th percentile URLs

but the values calculated using the Youden index, which can show optimal performance by combining sensitivity and specificity based on the ROC curve. The cut-offs obtained here showed similar values to the manufacturer's URL in the entire patient group, male, and over 50 years, and their diagnostic performance was also similar. In the patient group under 50 years of age, the specificity and PPV inevitably deteriorate because a low cut-off was calculated to maintain appropriate sensitivity.

 The 99th percentile URLs obtained from our previous study were from the healthy population who visit our institution for health check-upsEven though appropriate inclusion criteria and well-known outlier removal methods suggested in some guidelines were applied, but the 99th percentile URLs were lower than those from studies. Some studies for 99th percentile URL in Asians also showed similar findings, and the exact cause is still unknown. Because of the low URLsthe sensitivity and NPV are inevitably high, but specificity and PPV are low. And PPV and NPV are affected by disease prevalence. In our groups, women and patients under 50 years showed lower predictive values because the AMI prevalence in these group were < 7%. Some comments about the relationship with the prevalences were described in lines 312-313. 

Reviewer 2 Report

Dear Authors, I have read your manuscript with interest.

The current manuscript titled: "An algorithmic approach is superior to 99th percentile upper reference limits of high sensitivity troponin as a threshold for safe discharge from the emergency department" represents an important analysis of evolving field of Emergency Medicine and Cardiology.

In my opinion, these are the adjustments which should be made to increase the value of your manuscript:

  1. Please, write the article title with capital letters.
  2. Line 105: Check, please, if in the Emergency Department the cardiac exercise tests are performed for a patient with acute chest pain. If yes, please indicate which tests were performed, their indications and risks in emergency
  3. 2.2. Data collection and clinical assessment: For the differential diagnosis, whether other emergency diagnostic methods for acute chest pain, such as a CT angiography (i.g., to exclude aortic dissection), have been used? If yes, specify please.
  4. Table 1: Add, please, more baseline characteristics (i.g., BMI, arterial diseases, presence of heart failure [an increase in TnI can also be caused by heart failure presence] and its stages, smoking, dyslipidemia, chronic kidney disease, etc.).
  5. In Figures 1, 2 and 3, please change the font to match the one in the article text.
  6. Line 188, 195: Please, revise the term „stable angina” according to the new guideline – 2019 ESC Guidelines for the diagnosis and management of chronic coronary syndromes.
  7. Table 3: Change, please, word “Cutoffs” to “Cut-offs” and add in the table the column with p-values.
  8. For a better visual understanding, I advise you to add ROC curves.
  9. In the Discussions chapter, discuss the associated comorbidities that may increase TnI and how much these changes may influence the final diagnosis.
  10. Line 243: Figure 4 is missing from the text.
  11. To introduce your algorithm into real clinical practice, you need to provide internal and external validations. So, add this results please or discuss about this in Discussions chapter.
  12. Line 355 Conclusions: Please, exclude the aim from conclusions and add the practical significance of this study.
  13. References: Try to add new last guidelines (i.g. change reference 4 (guideline 2012), because is new 2020 ESC Guidelines for the management of acute coronary syndromes in patients presenting without persistent ST-segment elevation: The Task Force for the management of acute coronary syndromes in patients presenting without persistent ST-segment elevation of the European Society of Cardiology (ESC)).
  14. The manuscript contains some punctuation errors, please revise the text (i.g. try to homogenize the short and long dashes between numbers. Lines 124: 2-3 h and line 135: 0/2–3 h, etc.).

The Authors are to be congratulated on their work.

Good luck!

Author Response

Responses to the comments from the reviewers:

To reviewer 2
Thank you very much for your thoughtful suggestions and insights. The manuscript has benefited from these insightful suggestions. The manuscript has been rechecked and the necessary changes have been made in accordance with the reviewers’ suggestions. Changes in the main text have been highlighted in yellow.

 1. Please, write the article title with capital letters.

⇒ In the title sentence, the first letter of the word has been changed to a capital letter. 

2. Line 105: Check, please, if in the Emergency Department the cardiac exercise tests are performed for a patient with acute chest pain. If yes, please indicate which tests were performed, their indications and risks in emergency.

 ⇒ Thank you for your important comment. In our emergency department, cardiac exercise tests such as treadmill test, stress echocardiography or cardiac SPECT is not routinely performed. In this study, if the patients were categorized into Rule-In criteria or have typical chest pain with dynamic ECG change, we usually perform coronary angiography after admission. If not, the patients are followed up on an out-patients clinic, and if the cardiologist decides to differentiate ischemic heart disease, the above-mentioned functional test is performed. The phrase ‘exercise test’ has been deleted from this paragraph (2.2 Data collection and clinical assessment) to avoid confusion of the readers.

 3. 2.2. Data collection and clinical assessment: For the differential diagnosis, whether other emergency diagnostic methods for acute chest pain, such as a CT angiography (i.g., to exclude aortic dissection), have been used? If yes, specify please.

 ⇒ Most patients with chest pain (except STEMI) had an enhanced chest CT and in some patients (about 5%) who need to rule out aortic dissections and pulmonary embolisms, thoracic aorta CT angiography, pulmonary CT angiography was performed. So we added the phrases as below (in lines 106-107). 

 ‘In addition, examination findings such as vital signs, ECG, laboratory results, coronary angiography, echocardiography, and radiologic findings such as enhanced chest computed tomography (CT), thoracic aorta CT angiography, pulmonary CT angiography were collected.’

4. Table 1: Add, please, more baseline characteristics (i.g., BMI, arterial diseases, presence of heart failure [an increase in TnI can also be caused by heart failure presence] and its stages, smoking, dyslipidemia, chronic kidney disease, etc.).

⇒ We think your comments are right, but unfortunately, we did not collect information such as BMI, smoking, dyslipidemia, etc. The presence of ESRD was already described, and the frequencies of heart failure and cases with eGFR <60 were added in table 1.  

 5. In Figures 1, 2, and 3, please change the font to match the one in the article text.

⇒ All of the characters in the figures were changed into the same font of texts. 

6. Line 188, 195: Please, revise the term „stable angina” according to the new guideline – 2019 ESC Guidelines for the diagnosis and management of chronic coronary syndromes.

⇒ As you pointed out, we changed ‘stable angina’ to ‘chronic coronary syndromes.

7. Table 3: Change, please, the word “Cutoffs” to “Cut-offs” and add in the table the column with p-values.

⇒ We changed the “Cutoffs” to “Cut-offs” in the first row of table 3. Presenting the p-value in analyzing sensitivity, specificity, and predictive value does not seem to be a common research work. Instead, 95% CIs were added as it is common and could increase the objectivity of our data.

8. For a better visual understanding, I advise you to add ROC curves.

 We inserted ROC curves as figure 3.

9. In the Discussions chapter, discuss the associated comorbidities that may increase TnI and how much these changes may influence the final diagnosis.

⇒ In table 1, the history of heart failure and low eGFR were added and in the second paragraph of ‘Discussion’, some description for the risk factors and low eGFR was also added.

10. Line 243: Figure 4 is missing from the text.

⇒ ‘Figure 4’ in this sentence was in error and has been deleted.

11. To introduce your algorithm into real clinical practice, you need to provide internal and external validations. So, add this results please or discuss about this in Discussions chapter.

⇒ As you recommend, verification with a separate derivation and a validation cohort is ideal. Due to the limitation of time and available patients group, we could not validate with a new independent patients groups. So we used the internal validation method as you recommend. Total patients (N = 1,296) were divided 60:40 ratio along the sequence of ED visits and finally, the derivation and the validation cohort included 518 and 394 patients, respectively. Their results were added in the final parts of the ‘Results” section with figure 5-(A) and -(B). Results of each algorithm (total population, derivation cohort and validation cohort) were not significantly different. It was described in lines 266-272.

12. Line 355 Conclusions: Please, exclude the aim from conclusions and add the practical significance of this study.

⇒ In conclusion, the sentence corresponding to the aim was deleted and modified concisely.

13. References: Try to add new last guidelines (i.g. change reference 4 (guideline 2012), because is new 2020 ESC Guidelines for the management of acute coronary syndromes in patients presenting without persistent ST-segment elevation: The Task Force for the management of acute coronary syndromes in patients presenting without persistent ST-segment elevation of the European Society of Cardiology (ESC)).

⇒ We deleted the previous reference 4 and replaced it with the new ESC guideline 2020.  

14. The manuscript contains some punctuation errors, please revise the text (i.g. try to homogenize the short and long dashes between numbers. Lines 124: 2-3 h and line 135: 0/2–3 h, etc.).

⇒ All long dashes had been corrected to short dashes in the phrase meaning ‘ranges’. The manuscript has been thoroughly reviewed and corrected to eliminate other typos and punctuation errors.

Round 2

Reviewer 2 Report

I agree with the new submission.

Author Response

We added two sentences in the ‘Result’ section and corrected some sentences in ‘discussion’ section because a reviewer commented to discuss in the text about the low eGFR group. Because we added some sentences about the low eGFR group in the discussion section (the second and the second from the end paragraphs) in the previous revision, we did not add more sentences in the ‘discussion’.  Changes in the main text could be tracked using ‘Track Change” function in the MS word file of the revised manuscript.

- Added sentence in the ‘Result’ section:

“The frequencies of risk factors such as diabetes, hypertension, and heart failure were higher in the AMI group than in the non-AMI group, although it did not reach statistical significance. The frequencies of previous coronary artery diseases (37.4% vs 28.7%, P=0.0485) and lower estimated glomerular filtration rates (eGFR, <60 mL/min/1.73 m2) (5.8% vs 3.0%, P=0.0045) were significantly higher in the AMI patients than in the non-AMI patients.”

For our English editing, we have completed the English editing service for the first manuscript. English correction for the revised version was omitted, because the Medicina office answered that they would do its own English editing program before making the final proof reading version.